# How predictive of SARS-CoV-2 infection are clinical characteristics at presentation among individuals with COVID-like symptoms treated at the Mexican Institute of Social Security

Juan Pablo Gutierrez[1]*, Gustavo Olaiz[1], Arturo Juárez-Flores[1], Víctor H. Borja-Aburto[2], Iván J. Ascencio-Montiel[3], Stefano M. Bertozzi[4,5,6]

1 Center for Policy, Population and Health Research, School of Medicine, National Autonomous University of Mexico, Mexico City, Mexico, 2 Education and Research Unit, Mexican Institute of Social Security, Benito Juarez, Mexico City, Mexico, 3 Coordination of Epidemiological Surveillance, Mexican Institute of Social Security, Benito Juarez, Mexico City, Mexico, 4 University of California, Berkeley, California, United States of America, 5 University of Washington, Seattle, Washington, United States of America, 6 National Institute of Public Health, Mexico (INSP), Cuernavaca, Mexico

* jpgutierrez@unam.mx

**Data Availability Statement:** Access to the administrative data used for this analysis

## Abstract

### Background

The COVID-19 pandemic has progressed rapidly, with the emergence of new virus variants that pose challenges in treating infected individuals. In Mexico, four epidemic waves have been recorded with varying disease severity. To understand the heterogeneity in clinical presentation over time and the sensitivity and specificity of signs and symptoms in identifying COVID-19 cases, an analysis of the changes in the clinical presentation of the disease was conducted.

### Aim

To analyze the changes in the clinical presentation of COVID-19 among 3.38 million individuals tested for SARS-CoV-2 at the Mexican Social Security Institute (IMSS) from March 2020 to October 2021 and evaluate the predictivity of signs and symptoms in identifying COVID-19 cases.

### Methods

A retrospective analysis of clinical presentation patterns of COVID-19 among individuals treated at IMSS was performed, contrasting the signs and symptoms among SARS-CoV-2-positive individuals with those who tested negative for the virus but had respiratory infection symptoms. The sensitivity and specificity of each sign and symptom in identifying SARS-CoV-2 infection were estimated.

(SINOLAVE) is legally restricted by the IMSS law; authors have been granted access by a research agreement. Sharing of de-identified aggregated data will be considered by authors upon reasonable request. Data request may be also submitted to the Head of the Coordination of Epidemiological Surveillance, Mexican Institute of Social Security at xochitl.romero@imss.gob.mx.

**Funding:** This project was partially funded by a C3. ai Digital Transformation Institute grant (grant number N/A). There was no additional funding received for this study. The funders had no role in study design, data collection and analysis, decision to publish, or preparation of the manuscript.

**Competing interests:** All co-authors have seen and agree with the manuscript's contents and have no conflicts of interest to declare.

## Results

The set of signs and symptoms reported for COVID-19-suspected patients treated at IMSS were not highly specific for SARS-CoV-2 positivity. The signs and symptoms exhibited variability based on age and epidemic wave. The area under the receiver operating characteristic (ROC) curve was 0.62 when grouping the five main symptoms (headache, dyspnea, fever, arthralgia, and cough). Most of the individual symptoms had ROC values close to 0.5 (16 out of 22 between 0.48 and 0.52), indicating non-specificity.

## Conclusions

The results highlight the difficulty in making a clinical diagnosis of COVID-19 due to the lack of specificity of signs and symptoms. The variability of clinical presentation over time and among age groups highlights the need for further research to differentiate whether the changes are due to changes in the virus, who is becoming infected, or the population, particularly with respect to prior infection and vaccination status.

## Introduction

The COVID-19 pandemic has severely affected Mexico, with a high mortality rate due to both COVID-19 itself and the impact it has had on health services for other illnesses [1]. Responding to this infection has had to occur on the fly, reflecting experience accumulated over time. Adjustments made have reflected advances in knowledge about the virus and changes in the virus itself, including the emergence of new dominant variants with different infectivity and severity [2, 3].

One of the challenges in any new pandemic is the inability to diagnose people in a timely manner, as discussed in other studies [4], particularly in low- and middle-income countries where testing constraints due to excess demand and limited testing supply are even more pronounced than in high-income countries [5].

Overwhelmed health services—especially hospital services—have been identified as related to worse outcomes as the quality-of-care decrease with evidence of increased mortality [6, 7]. During the COVID-19 pandemic, the quality of care has decreased [8, 9] due to exceeding capacity but also because of staff burnout.

The use of pulmonary radiographs has been proposed as a mechanism for timely diagnosis, given its relatively low cost, use of widely installed infrastructure, and immediate results. However, its low specificity for differentiating between COVID-19 and other respiratory infections limits its potential [10]. The combined use of different diagnostic tools, such as radiographs combined with signs and symptoms, has been suggested, but these approaches have not adequately substituted for laboratory tests for SARS-CoV-2 [11, 12].

A potential approach is to use the clinical characteristics of patients at presentation to identify those who are most likely to be SARS-CoV-2 positive. This approach has been implemented with alternative definitions of what constitutes a likely case and/or severe case of COVID-19 from the presenting signs and symptoms. While it is clear that an individual with respiratory stress requires immediate care, it is still useful to discern if the individual is affected by SARS-CoV-2 or other respiratory conditions.

Nevertheless, while different signs and symptoms have been proposed as indicative of COVID-19, studies across the globe have identified a large list of signs and symptoms among individuals later confirmed as COVID-19 cases. Among the alternative profiles, the Mexican

government defined a likely case and likely severe case based on a set of signs and symptoms at presentation [13].

Identifying the most likely profile of signs and symptoms among individuals with COVID-19-like symptoms could contribute to timely care and—especially among those resulting in severe cases—decrease risk of death.

With the advantage of a very large dataset of individuals suspected of COVID-19 due to respiratory symptoms, our aim was to predict SARS-CoV-2 positivity using reported patient-level demographic characteristics and signs and symptoms, taking into account changes in patterns over time through three different epidemic waves.

## Methods

We conducted a cross-sectional analysis of data from the IMSS Epidemiological Surveillance Online Notification System (SINOLAVE). As previously reported [5], SINOLAVE served as the platform for recording suspected COVID-19 cases within IMSS. The responsible epidemiology personnel in each of the 1,515 first-level medical units, 248 second-level care hospitals, and 10 national medical centers validated and recorded the data from the epidemiological studies, diagnostic test results, hospitalization data, and, if applicable, death of COVID-19 suspected patients. Upon presentation, data were collected from the patients or their relatives and included demographic and clinical information, with almost no missing data.

To analyze the evolution of symptomatology, we divided the Mexican pandemic into three waves: February to September 2020, October 2020 to May 2021, and April to October 2021. The initiation of a new wave was identified as the lowest point in case numbers subsequent to the preceding wave.

We only analyzed suspected cases with valid PCR or rapid test results to assess the sensitivity and specificity of signs and symptoms. Positive cases were defined as those with a positive result in either of these two tests, while negative cases had only negative results.

We utilized the Ministry of Health (Secretaria de Salud) definitions for suspect and severe cases. A suspect case was defined as a patient presenting with at least one major symptom (cough, dyspnea, fever, or headache) and one minor symptom (myalgias, arthralgias, odynophagia, chills, chest pain, rhinorrhea, polypnea, anosmia, dysgeusia, or conjunctivitis). Severe cases were defined as suspect cases who also exhibited dyspnea or chest pain [13].

### Analysis

In this analysis, the results of rapid antigen tests were considered equivalent to those of PCR tests, disregarding the possibility of false positives. As such, individuals with either a positive PCR or rapid test result were classified as positive cases. To determine the sensitivity and specificity of symptoms and signs in predicting infection, we compared their prevalence among positive and negative test results in Stata. Sensitivity was defined as the proportion of positive cases with a specific symptom to the total number of positive cases, while specificity was defined as the proportion of negative cases without the symptom to the total number of negative cases; this was calculated as:

$$Sensitivity_i = \frac{Positive\ cases\ with\ symptom_i}{Positive\ Cases}.$$

$$Specificity_i = \frac{Negative\ cases\ without\ symptom_i}{Negative\ Cases}$$

Using these values, we generated a Receiver Operating Characteristic (ROC) curve by plotting sensitivity against 1-specificity, and calculated the area under the curve (AUC) using a non-parametric approach in Stata. An AUC of 0.5 represents a symptom that is not informative in relation to the presence of COVID-19, while an AUC of 1 indicates a symptom that perfectly identifies the presence of COVID-19. A value of 0.75 or greater is considered a good predictor [14, 15].

Additionally, considering that age has been a key predictor of severity among COVID-19 patients, we estimated the ROC AUC by age group for individuals aged 0 to 18 years, 19 to 49 years, and 50 years and older.

## Ethics

This analysis was conducted as part of a project that was reviewed and approved by the Research and Ethics Committee (IRB) of the Mexican Social Security Institute (IMSS) (registration number R-2020-785-165). The study was approved with the waiver of informed consent, and all data was anonymized. The procedures and methods followed IMSS research guidelines and national norms for health research.

## Results

From a total of 4.48 million people seeking care for possible COVID-19 symptoms, 3.38 million underwent either a PCR or rapid test. Only tested individuals were included in this analysis. Table 1 displays the prevalence of symptoms by wave and SARS-CoV-2 test result among the Mexican population treated at IMSS facilities. As shown, there are significant differences between those with positive and negative SARS-CoV-2 test results for symptoms such as fever, cough, arthralgia, myalgia, and dyspnea, with differences reaching up to 17 percentage points. For most symptoms, the prevalence among those with positive test results is similar to those with negative test results, with differences of less than 5 percentage points. There are also notable differences in the prevalence of symptoms between waves: for instance, the prevalence of fever, which was 77.5% among those with a positive test during the first wave, decreased to 65.2% during the third wave. Additionally, the difference in prevalence between those with positive and negative test results also changed between waves: for example, the prevalence of dyspnea was 14.9 percentage points higher among SARS-CoV-2 positive patients compared to those with negative tests during the first wave, but only 4.6 percentage points higher during the third wave.

The sensitivity and specificity of each symptom for predicting COVID-19 infection are reported in Table 2, which includes the prevalence and the area under the receiver operating characteristic curve (AUC ROC). The results reveal that none of the analyzed symptoms have an AUC ROC of 0.75 or higher, and most have a value close to 0.5, indicating limited usefulness as predictors. Fever, anosmia, dysgeusia, and cough have the highest AUC ROC, ranging from 0.53 to 0.58 in the three waves. Most of the other symptoms have an AUC ROC around 0.50.

The definition of a suspected case has an AUC ROC of 0.56 during wave 1, and 0.57 for waves 2 and 3, which are similar to the values for presenting with fever, anosmia, dysgeusia, or cough individually. The definition of a severe case also produces a similar AUC ROC: 0.56 in wave 1, 0.57 in wave 2, and 0.53 in wave 3. The combination of the five most frequently reported COVID-19 symptoms (dyspnea, fever, cough, arthralgias, and headache) as a categorical variable produces the highest AUC ROC of 0.62 in the three waves. Adding anosmia increases the AUC ROC slightly to 0.64, still indicating limited predictive power.

**Table 1. Prevalence (observations) of signs and symptoms among suspected cases by confirmed COVID-19 status.**

| Symptom | Wave 1 | | | Wave 2 | | | Wave 3 | | |
|---|---|---|---|---|---|---|---|---|---|
| | Negative | Positive | Difference (percent points) | Negative | Positive | Difference (percent points) | Negative | Positive | Difference (percent points) |
| FEVER | 63.0% (82,872) | 77.5% (129,076) | 14.6*** | 47.8% (342,722) | 61.9% (313,496) | 14.1*** | 48.4% (565,329) | 65.2% (449,973) | 16.8*** |
| ANOSMIA | 5.9% (7,740) | 11.8% (19,605) | 6.0*** | 11.5% (82,7129) | 25.6% (129,622) | 14.1*** | 8.3% (96,956) | 23.2% (160,133) | 14.9*** |
| DYSGEUSIA | 5.9% (7,739) | 11.2% (18,679) | 5.3*** | 11.2% (80,622) | 23.7% (120,285) | 12.5*** | 7.9% (91,516) | 21.3% (147,045) | 13.4*** |
| COUGH | 68.3% (89,935) | 80.4% 133,807) | 12.1*** | 62.0% (444,984) | 75.0% (380,283) | 13.0*** | 63.4% (740,369) | 76.2% (525,949) | 12.9*** |
| ARTHRALGIA | 51.2% (67,354) | 60.7% (100,946) | 9.5*** | 45.1% (323,355) | 55.3% (280,337) | 10.3*** | 41.5% (485,392) | 51.6% (355,834) | 10.0*** |
| MYALGIA | 56.7% (74,611) | 66.0% (109,755) | 9.2*** | 52.2% (374,916) | 61.6% (312,109) | 9.4*** | 49.5% (578,866) | 58.8% (405,753) | 9.3*** |
| DYSPNEA | 25.7% (33,773) | 40.6% 67,560) | 14.9*** | 14.4% (103,574) | 29.0% (147,146) | 14.6*** | 9.3% (108,938) | 14.0% (96,327) | 4.6*** |
| CHILLS | 34.1% (44,808) | 40.1% (66,813) | 6.1*** | 31.7% (227,449) | 39.5% (200,386) | 7.9*** | 26.6% (310,984) | 31.1% (214,346) | 4.5*** |
| MALAISE | 46.9% (61,668) | 54.3% (90,336) | 7.4*** | 33.7% (241,681) | 43.8% (221,876) | 10.1*** | 30.0% (350,860) | 34.1% (235,422) | 4.1*** |
| HEADACHE | 78.5% (103,304) | 80.3% (133,683) | 1.8* | 75.3% (540,091) | 77.6% (393,191) | 2.3** | 75.3% (879,433) | 78.5% (541,555) | 3.2*** |
| THORACIC PAIN | 25.5% (33,537) | 31.7% (52,726) | 6.2*** | 17.8% (127,897) | 24.7% (125,073) | 6.9*** | 13.1% (152,609) | 15.1% (104,071) | 2.0*** |
| RHINORRHEA | 27.6% (36,287) | 28.6% (47,536) | 1.0 | 41.0% (293,987) | 38.1% (192,971) | -2.9** | 45.6% (532,995) | 47.5% (327,593) | 1.9** |
| ODYNOPHAGIA | 47.1% (61,912) | 48.7% (81,115) | 1.7 | 54.2% (388,711) | 53.1% (269,177) | -1.0 | 57.3% (669,713) | 58.5% (403,471) | 1.2* |
| POSTRATION | 4.5% (5,957) | 5.9% (9,762) | 1.3*** | 2.9% (21,063) | 5.5% (27,864) | 2.6*** | 2.2% (25,393) | 3.1% (21,323) | 0.9*** |
| CYANOSIS | 2.3% (3,046) | 3.6% (6,065) | 1.3*** | 1.0% (7,174) | 2.6% (13,038) | 1.6*** | 0.6% (6,497) | 0.9% (6,268) | 0.4*** |
| POLYPNEA | 2.3% (3,046) | 3.6% (6,065) | 1.3*** | 1.0% (7,174) | 2.6% (13,038) | 1.6*** | 0.6% (6,497) | 0.9% (6,268) | 0.4*** |
| CONJUNTIVITIS | 8.9% (11,743) | 8.2% (13,623) | -0.7* | 7.7% (55,466) | 7.9% (39,935) | 0.2 | 7.1% (83,019) | 7.4% (51,234) | 0.3 |
| CORYZA | 2.0% (2,577) | 2.4% (4,050) | 0.5* | 1.4% (10,356) | 2.3% (11,573) | 0.8*** | 1.1% (13,343) | 1.5% (10,034) | 0.3** |
| OTHER | 3.1% (4,115) | 3.5% (5,788) | 0.4* | 2.1% (14,842) | 2.5% (12,512) | 0.4**** | 1.6% (18,489) | 1.7% (11,978) | 0.2 |
| IRRITABILITY | 13.5% (17,699( | 14.5% (24,185) | 1.1 | 8.8% (63,359) | 10.2% (51,860) | 1.4* | 8.7% (101,497) | 8.6% (59,401) | -0.1* |
| ABDOMINAL PAIN | 15.3% (20,182) | 13.9% (23,148) | -1.4** | 10.9% (78,110) | 9.9% (50,185) | -1.0** | 10.1% (117,448) | 7.7% (53,265) | -2.3*** |
| DIARRHEA | 22.6% (29,795) | 21.9% (36,367) | -0.8 | 16.3% (117,065) | 14.5% (73,450) | -1.8*** | 15.3% (178,759) | 11.5% (79,374) | -3.8*** |
| Observations | 131,588 | 166,426 | | 717,726 | 506,766 | | 1,168,411 | 689,860 | |

*p<0.05

**p<0.01

***p<0.001

**Table 2. Sensitivity (Sens.), specificity (Spec.), prevalence (%), and area under the ROC curve (ROC AUC) of signs and symptoms for confirmed COVID-19.**

| | Wave 1 | | | | Wave 2 | | | | Wave 3 | | | |
|---|---|---|---|---|---|---|---|---|---|---|---|---|
| | Sens. | Spec. | % (95% CI) | ROC AUC (95% CI) | Sens. | Spec. | % (95% CI) | ROC AUC (95% CI) | Sens. | Spec. | % (95% CI) | ROC AUC (95% CI) |
| Main 5* | | | | 0.62 (0.62–0.62) | | | | 0.62 (0.62–0.62) | | | | 0.62 (0.62–0.62) |
| Main 6** | | | | 0.63 (0.63–0.63) | | | | 0.64 (0.64–0.64) | | | | 0.64 (0.64–0.64) |
| Fever | 60.9% | 56.6% | 71.1 (68.4–73.8) | 0.57 (0.57–0.57) | 47.8% | 66.0% | 53.6 (51.1–56.0) | 0.57 (0.57–0.57) | 44.3% | 71.5% | 54.6 (51.9–57.4) | 0.58 (0.58–0.58) |
| Anosmia | 71.7% | 45.8% | 9.2 (7.8–10.6) | 0.53 (0.53–0.53) | 61.0% | 62.7% | 17.3 (15.4–19.2) | 0.57 (0.57–0.57) | 62.3% | 66.9% | 13.8 (12.3–15.4) | 0.57 (0.57–0.58) |
| Suspected case^ | 60.1% | 52.9% | 67.2 (63.0–71.3) | 0.56 (0.56–0.56) | 46.8% | 66.5% | 59.5 (56.2–62.7) | 0.57 (0.57–0.57) | 42.9% | 70.7% | 57.3 (54.3–60.3) | 0.57 (0.57–0.57) |
| Dysgeusia | 70.7% | 45.6% | 8.9 (7.5–10.3) | 0.53 (0.53–0.53) | 59.9% | 62.2% | 16.4 (14.6–18.3) | 0.56 (0.56–0.56) | 61.4% | 66.5% | 12.9 (11.4–14.4) | 0.57 0.57–0.57) |
| Cough | 59.8% | 56.1% | 75.1 (72.6–77.5) | 0.56 (0.56–0.56) | 46.1% | 68.3% | 67.4 (65.0–69.8) | 0.57 (0.56–0.57) | 41.5% | 72.3% | 68.1 (66.1-70-2) | 0.56 (0.56–0.57) |
| Arthralgias | 60.0% | 49.5% | 56.5 (52.9–60.1) | 0.55 (0.55–0.55) | 46.4% | 63.5% | 49.3 (46.8–51.8) | 0.55 0.55–0.55) | 42.3% | 67.2% | 45.3 (42.8–47.8) | 0.55 (0.55–0.55) |
| Myalgias | 59.5% | 50.1% | 61.9 (58.4–65.4) | 0.55 (0.54–0.55) | 45.4% | 63.8% | 56.1 (53.7–58.5) | 0.55 0.55–0.55) | 41.2% | 67.5% | 53.0 (50.9–55.1) | 0.55 (0.55–0.55) |
| Dyspnea | 66.7% | 49.7% | 34.0 (30.7–37.3) | 0.57 (0.57–0.58) | 58.7% | 63.1% | 20.5 (17.9–23.1) | 0.57 (0.57–0.57) | 46.9% | 64.1% | 11.0 (9.3–12.8) | 0.52 (0.52–0.52) |
| Chills | 59.9% | 46.6% | 37.5 (33.6–41.3) | 0.53 (0.53–0.53) | 46.8% | 61.5% | 34.9 (31.7–38.2) | 0.54 (0.54–0.54) | 40.8% | 64.3% | 28.3 (24.5–32.0) | 0.52 (0.52–0.52) |
| Malaise | 59.4% | 47.9% | 51.0 (47.3–54.7) | 0.54 (0.54–0.54) | 47.9% | 62.6% | 37.9 (34.6–41.2) | 0.55 (0.55–0.55) | 40.2% | 64.3% | 31.5 (28.7–34.4) | 0.52 (0.52–0.52) |
| Headache | 56.4% | 46.4% | 79.5 (77.4–81.7) | 0.51 (0.51–0.51) | 42.1% | 61.0% | 76.2 (74.3–78.2) | 0.51 (0.51–0.51) | 38.1% | 66.1% | 76.5 (74.9–78.0) | 0.52 (0.52–0.52) |
| Chest pain | 61.1% | 46.3% | 28.9 (26.3–31.6) | 0.53 (0.53–0.53) | 49.4% | 60.7% | 20.7 (18.4–22.9) | 0.53 (0.53–0.54) | 40.5% | 63.4% | 13.8 (11.8–15.8) | 0.51 (0.51–0.51) |
| Rhinorrhea | 56.7% | 44.5% | 28.1 (26.1–30.2) | 0.50 (0.50–0.51) | 39.6% | 57.5% | 39.8 (37.0–42.5) | 0.49 (0.48–0.49) | 38.1% | 63.7% | 46.3 (43.7–48.9) | 0.51 (0.51–0.51) |
| Odynophagia | 56.7% | 45.0% | 48.0 (45.5–50.5) | 0.51 (0.51–0.51) | 40.9% | 58.1% | 53.7 (50.8–56.7) | 0.49 (0.49–0.50) | 37.6% | 63.5% | 57.8 (55.5–60.0)- | 0.51 (0.51–0.51) |
| Prostration | 62.1% | 44.5% | 5.3 (3.7–6.9) | 0.51 (0.51–0.51) | 57.0% | 59.3% | 4.0 (2.9–5.1) | 0.51 (0.51–0.51) | 45.6% | 63.1% | 2.5 (1.7-3-3) | 0.50 (0.50–0.50) |
| Cyanosis | 66.6% | 44.5% | 3.1 (2.3–3.8) | 0.51 (0.51–0.51) | 64.5% | 59.0% | 1.7 (1.1–2.2) | 0.51 (0.51–0.51) | 49.1% | 63.0% | 0.7 (0.5–0.9) | 0.50 (0.50–0.50) |
| Polypnea | 66.6% | 44.5% | 3.1 (2.3–3.8) | 0.51 (0.51–0.51) | 64.5% | 59.0% | 1.7 (1.1–2.2) | 0.51 (0.51–0.51) | 49.1% | 63.0% | 0.7 (0.4–0.9) | 0.50 (0.50–0.50) |
| Coryza | 61.1% | 44.3% | 2.2 (1.5–2.9) | 0.50 (0.50–0.50) | 52.8% | 58.8% | 1.8 (1.3–2.3) | 0.50 (0.50–0.50) | 42.9% | 63.0% | 1.3 (0.8–1.6) | 0.50 (0.50–0.50) |
| Conjunctivitis | 53.7% | 44.0% | 8.5 (7.1–9.9) | 0.50 (0.50–0.50) | 41.9% | 58.7% | 7.8 (6.5–9.1) | 0.50 0.50–0.50) | 38.2% | 63.0% | 7.2 (5.9–8.6) | 0.50 (0.50–0.50) |
| Other | 58.4% | 44.2% | 3.3 (2.7–4.0) | 0.50 (0.50–0.50) | 45.7% | 58.7% | 2.2 (1.8–2.7) | 0.50 (0.50–0.50) | 39.3% | 62.9% | 1.6 (1.4–1.9) | 0.50 (0.50–0.50) |
| Abdominal pain | 53.4% | 43.7% | 14.5 (12.7–16.4) | 0.49 (0.49–0.49) | 39.1% | 58.3% | 10.5 (9.1–11.9) | 0.50 (0.49–0.50) | 31.2% | 62.3% | 9.2 (7.9–10.5) | 0.49 (0.49–0.49) |
| Irritability | 57.7% | 44.5% | 10.3% | 0.51 (0.49–0.53) | 45.0% | 59.0% | 7.6% | 0.48 (0.47–0.50) | 36.9% | 62.9% | 8.5% | 0.49 (0.47–0.50) |
| Diarrhea | 55.0% | 43.9% | 22.2 (20.5–23.9) | 0.50 (0.49–0.50) | 38.6% | 58.1% | 15.6 (14.1–17.0) | 0.49 (0.49–0.49) | 30.7% | 61.8% | 13.9 (12.9–14.9) | 0.48 (0.48–0.48) |
| Severe case^^ | 64.7% | 49.1% | 35.7 (32.2–39.1) | 0.56 (0.56–0.57) | 55.2% | 62.9% | 23.5 (20.6–26.4) | 0.57 0.57–0.57) | 46.0% | 64.4% | 14.9 (12.6–17.3) | 0.53 (0.53–0.53) |

*Five most common: dyspnea, fever, cough, arthralgias, and headache

**Six most common: adding anosmia to the five before

^Suspect case: a patient presenting with at least one major symptom (cough, dyspnea, fever, or headache) and one minor symptom (myalgias, arthralgias, odynophagia, chills, chest pain, rhinorrhea, polypnea, anosmia, dysgeusia, or conjunctivitis)

^^Severe cases were defined as suspect cases who also exhibited dyspnea or chest pain

**Table 3. Area under the ROC curve [ROC AUC] (95% confidence interval) of categorical variables with five and six signs and symptoms, by age group.**

| Age group | Wave 1 | | Wave 2 | | Wave 3 | |
|---|---|---|---|---|---|---|
| | Main 5 | Main 6 | Main 5 | Main 6 | Main 5 | Main 6 |
| 0 to 18 | 0.57 (0.56–0.59) n = 7470 | 0.58 (0.57–0.60) | 0.59 (0.58–0.59) n = 47948 | 0.61 (0.61–0.62) | 0.61 (0.60–0.61) n = 179647 | 0.62 (0.62–0.62) |
| 19 to 49 | 0.60 (0.60–0.61) n = 189038 | 0.62 (0.61–0.62) | 0.59 (0.59–0.59) n = 847120 | 0.62 (0.62–0.62) | 0.61 (0.61–0.61) n = 1359824 | 0.63 (0.63–0.64) |
| 50+ | 0.62 (0.62–0.62) n = 101572 | 0.62 (0.62–0.63) | 0.67 (0.66–0.67) N = 329507 | 0.67 (0.67–0.67) | 0.64 (0.64–0.64) n = 318841 | 0.66 (0.66–0.66) |

As seen in Table 2, while the frequency of symptoms and their sensitivity and specificity change across the three waves, the values of the AUC ROC remain relatively stable, suggesting that losses in sensitivity are compensated by gains in specificity and vice versa.

Table 3 shows that while there is an increase in the AUC ROC for older individuals, even they have a maximum value of 0.67. For younger people (0–18 years), the AUC ROC for the categorical variable with the five most frequently reported symptoms is 0.57 in wave 1, 0.59 in wave 2, and 0.60 in wave 3, while adding anosmia increases these values to 0.58, 0.61, and 0.62, respectively. For those aged 19–49 years, the AUC ROC for the five symptoms is 0.60 in wave 1, 0.59 in wave 2, and 0.61 in wave 3, and increases to 0.62, 0.62, and 0.63, respectively, with the addition of anosmia. Finally, for those 50 years and older, the AUC ROC for the five symptoms is 0.62, 0.67, and 0.64 for waves 1, 2, and 3, respectively, and remains the same in waves 1 and 2 and increases to 0.66 in wave 3 with the addition of anosmia.

## Discussion

In line with studies conducted in various contexts, the symptomatology of COVID-19 among individuals treated at IMSS in Mexico during the first three waves of the pandemic (March 2020 to October 2021) has been non-specific, posing challenges for empirical diagnosis [16–19]. Our analysis indicates that none of the clinical characteristics, including signs and symptoms collected at presentation, exhibit a clinically relevant area under the receiver operating characteristic curve (AUC ROC) of 0.75 or higher. Even when combined, the highest achieved AUC ROC value is 0.62, slightly improving to 0.67 for those aged 50 and older. Notably, characteristics defined as severe cases in the Mexican guidelines only yield AUC ROC values between 0.53 and 0.57. Consequently, the clinical profile for both suspect and severe cases outlined in the guidelines demonstrates limited predictive capacity for SARS-CoV-2 positivity.

Among the 3 million individuals identified as COVID-19 suspects tested at IMSS, a majority exhibited similar symptom patterns. While the prevalence of symptoms was higher among those testing positive for SARS-CoV-2, the differences between positive and negative cases were not significant enough for clear differentiation.

Our analysis aligns with previous studies suggesting that individual symptoms alone or in combination have limited utility for SARS-CoV-2 detection [20, 21]. A study of cases in Ghana similarly found AUC ROC around 0.5 for individual symptoms, consistent with our results [22].

From a public policy perspective, widespread testing is crucial for effective population screening to control virus spread, especially as symptoms overlap with those of other viral infections. However, declining severity and pandemic fatigue have led to decreased demand for testing. The evolving nature of the virus further complicates tracking variations in clinical presentation, as evidenced by the three waves analyzed in Mexico.

The symptom profile of COVID-19 has changed over time, with varying prevalence between waves, and individuals are more likely to acquire immunity. Notably, individuals in the dataset identified as suspect cases did not universally meet Mexico's official suspect case definition, indicating varied criteria used by healthcare professionals. Additionally, only a small percentage of those meeting suspect or severe case profiles tested positive for SARS-CoV-2 (46% and 53%, respectively).

This study has limitations, including potential reporting bias in signs and symptoms at presentation. However, there's no indication that this bias varies between waves or test results. The data's quality, received from SINOLAVE, is beyond our control and may vary among facilities.

## Conclusion

It is crucial to recognize the need for an adaptable definition of a suspected COVID-19 case, considering the evolving profile of cases with each wave or variant of SARS-CoV-2. Our research highlights the inefficiency of maintaining a static list of signs and symptoms to predict the likelihood of infection. Even if the definition were refined to better predict positive cases retrospectively, there is no guarantee that it would remain relevant for future waves or variants with different symptom profiles. The identification of COVID-19 through symptoms alone is limited, although the presence of the five most common symptoms does suggest a higher probability of infection with SARS-CoV-2.

## Author Contributions

**Conceptualization:** Juan Pablo Gutierrez, Gustavo Olaiz, Iván J. Ascencio-Montiel, Stefano M. Bertozzi.

**Formal analysis:** Juan Pablo Gutierrez, Arturo Juárez-Flores.

**Funding acquisition:** Stefano M. Bertozzi.

**Writing – original draft:** Juan Pablo Gutierrez, Stefano M. Bertozzi.

**Writing – review & editing:** Juan Pablo Gutierrez, Gustavo Olaiz, Víctor H. Borja-Aburto, Iván J. Ascencio-Montiel, Stefano M. Bertozzi.

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
