## [Decision Letter · Decision Letter 0]

8 Jan 2023

PONE-D-22-14771Evolution of COVID-19 symptomatology in individuals treated at the Mexican Social Security Institute.PLOS ONE

Dear Dr. Gutierrez,

Thank you for submitting your manuscript to PLOS ONE. After careful consideration, we feel that it has merit but does not fully meet PLOS ONE’s publication criteria as it currently stands. Therefore, we invite you to submit a revised version of the manuscript that addresses the points raised during the review process.

Please address all concerns by both reviewers, especially those by Reviewer 2.

We look forward to receiving your revised manuscript.

Kind regards,

Siew Ann Cheong, Ph.D.

Academic Editor

PLOS ONE

Journal Requirements:

“This project was partially funded by a C3.ai Digital Transformation Institute grant (grant number N/A). The funders had no role in study design, data collection and analysis, decision to publish, or preparation of the manuscript.”

Reviewers' comments:

Reviewer's Responses to Questions

**Comments to the Author**

1. Is the manuscript technically sound, and do the data support the conclusions?

Reviewer #1: Partly

Reviewer #2: No

2. Has the statistical analysis been performed appropriately and rigorously? 

Reviewer #1: Yes

Reviewer #2: No

3. Have the authors made all data underlying the findings in their manuscript fully available?

Reviewer #1: No

Reviewer #2: No

4. Is the manuscript presented in an intelligible fashion and written in standard English?

Reviewer #1: No

Reviewer #2: Yes

5. Review Comments to the Author

Reviewer #1: This is a retrospective analysis of 3.4 million patients who were tested for SARS-CoV-2, with the aim to ascertain if clinical presentation may provide an accurate means to determine whether the patients is infected or not. The authors conclude the ability of symptoms to discern positive/negative results is generally poor.

Major Points:

As the title suggests, the intent is to show how the pandemic evolved over time. Is it perhaps more informative that a comparison within result status be performed? For example, did the prevalence of fever among COVID-positive change between the 3 waves? Some of the most common symptoms do vary between test result and age group, however these prevalences appear to be relatively similar across waves. This is more apparent in Figure 1 as compared to Table 1. It is interesting, particularly among ambulatory, that some of the biggest prevalence changes occur among the negatives? In that regard, are the difference changes listed in Table 1 driven more by evolution of COVID-negative or COVID-positive?

Figure 1 is cumbersome to view

The authors describe how clinical presentation varies by patient age as well as by location type (i.e., ambulatory vs. hospitalized). It would be worth noting whether the distribution of these two variables changed between Waves. Presumably they do but it may help the reader better understand what may be driving changes over time. Relatedly, there is no formal definition of ambulatory vs. hospitalized. This may be especially relevant if testing occurred in an ambulatory setting but then there was an escalation of care.

As Figure 1 suggests, the ROC values may show greater discerning ability for select age groups, rather than the overall population. Was there an examination of ROC by select age groups?

Minor Points:

The distinction between suspect and severe are somewhat blurred since dyspnea and chest pain are included in both definitions.

It’s unclear why some of the prevalences in Table 2 are discrepant from Table 1. For example, the lowest proportion for headache in Table #1 is 75.3% but the proportions for Wave 1 and Wave 2 are roughly 65%.

Line 76 states that only suspected [or severe] cases were included in the analysis. Consequently, examining the relationship between suspected case status and positive/negative result (Line 138) doesn’t seem clear.

Referencing the IMSS is helpful but the authors should consider adding an additional sentence describing how symptom data are collected. This will help the reader better understand the accuracy of select symptoms that may be hard to document in particularly young patients.

There are typos that need to be corrected.

Reviewer #2: The title of this article claims that the aim of the study is to describe the evolution of COVID-19 symptomatology in individuals treated at the Mexican Social Security Institute (IMSS). When reading the title we expect an analysis of the evolution of the symptoms over time per individual, which would be of high interest.

However when reading in details the abstract and the introduction the aim of the study appears not clearly: in the abstract it is mentioned that the aim is “to analyse the changes in the clinical presentation of COVID-19 among 3.38 million individuals tested at IMSS from march 2020 to October 2021” whereas in the introduction lines 59-60 the authors says “We aimed to predict SARS-COV2 positivity using reported patient-level demographic characteristics, signs and symptoms…”. This aims seems to be the real aim of the article. The manuscript is easy to read but lacks of depth. The methods are not detailed enough and discussion is very limited, there is no mention of the strengths and limitations of the study and there is no conclusion in the main text.

So at this stage I recommend rejecting this article.

Comments

I would recommend to the authors to improve the manuscript by clarifying its main objective and message and by rewording the title accordingly.

Lines 62-63: the last sentence should be removed and placed in the discussion and/or in the conclusion instead

Methodology: the methods should more detailed. There is no mention of how missing values were treated. This should be described here. Table 1: Add N for each category + add p-values

Discussion: Add a short summary of the findings of the study and an analysis of its strength and weaknesses.

Conclusions: there is no conclusion in the main text

Minor comments

There are some typing errors that should be corrected (ex line 102)

Line 107: Authors wrote : “Figure 1 and 2….” but there is only one figure in the document (Figure 1). On this figure, the group age are not readable. The police should be increased.

6. PLOS authors have the option to publish the peer review history of their article (what does this mean?). If published, this will include your full peer review and any attached files.

Reviewer #1: No

Reviewer #2: No

---

## [Author Response · Author response to Decision Letter 0]

14 Feb 2023

Please see the attached file names Response to reviewers.

---

## [Decision Letter · Decision Letter 1]

26 May 2023

PONE-D-22-14771R1How predictive of SARS-CoV-2 infection are clinical characteristics at presentation among individuals with COVID-like symptoms treated at the Mexican Institute of Social Security.PLOS ONE

Dear Dr. Gutierrez,

Thank you for submitting your manuscript to PLOS ONE. After careful consideration, we feel that it has merit but does not fully meet PLOS ONE’s publication criteria as it currently stands. Therefore, we invite you to submit a revised version of the manuscript that addresses the points raised during the review process.

While Reviewer 1 has recommended further minor changes, Reviewer 2 has declined to comment on the revised manuscript. Therefore, I would like to suggest to the authors: while you are responding to the comments of Reviewer 1, please look through the original comments of Reviewer 2, and respond to them as best as you can. I will invite a new reviewer to report on the new revision.

We look forward to receiving your revised manuscript.

Kind regards,

Siew Ann Cheong, Ph.D.

Academic Editor

PLOS ONE

Reviewers' comments:

Reviewer's Responses to Questions

**Comments to the Author**

1. If the authors have adequately addressed your comments raised in a previous round of review and you feel that this manuscript is now acceptable for publication, you may indicate that here to bypass the “Comments to the Author” section, enter your conflict of interest statement in the “Confidential to Editor” section, and submit your "Accept" recommendation.

Reviewer #1: (No Response)

2. Is the manuscript technically sound, and do the data support the conclusions?

Reviewer #1: Partly

3. Has the statistical analysis been performed appropriately and rigorously? 

Reviewer #1: No

4. Have the authors made all data underlying the findings in their manuscript fully available?

Reviewer #1: Yes

5. Is the manuscript presented in an intelligible fashion and written in standard English?

Reviewer #1: Yes

6. Review Comments to the Author

Reviewer #1: I thank the authors for the effort addressing reviewer commends. The authors have strengthened the manuscript by focusing the specific aim, removing some unnecessary complexities, and modifying the title.

Additional detail on IMSS has been added. However, I believe the paper could be strengthened further by including additional descriptions. For example, are there any validation or reliability statistic that IMSS has calculated? Something to assure the reader that medical data were abstracted in a reliable manner. This become more relevant, perhaps, when trying to understand how accurately some of these symptoms could be assessed among infants (e.g., headache, anosmia).

A level of precision is needed, be it confidence intervals (preferable) or p-values. The authors do have a large dataset. These data, however, are still a sample, as the authors mention:

Line 120: “From a total of 4.48 million people seeking care for possible COVID symptoms, 3.38 million underwent [testing].

This coupled with the statement in the discussion stating demand for testing changed over time suggests the possibility of selection bias. Where I think the level of precision becomes most imperative is when we are comparing prevalences/AUC between waves or between age groups. Having a confidence interval, however tight it may be, helps the reader understand what changes may be considered meaningful changes. Relatedly, there are no sample sizes in Table 3 to understand the frequency contribution of the 3 age groups.

Minor points:

Analysis describes a subgroup-analysis to compare positive non-severe and positive severe, but the reviewer was unable to find those results.

Why is anosmia included in the “main 6” but it’s not listed in Table 1?

7. PLOS authors have the option to publish the peer review history of their article (what does this mean?). If published, this will include your full peer review and any attached files.

Reviewer #1: No

---

## [Decision Letter · Decision Letter 2]

29 Nov 2023

PONE-D-22-14771R2How predictive of SARS-CoV-2 infection are clinical characteristics at presentation among individuals with COVID-like symptoms treated at the Mexican Institute of Social Security.PLOS ONE

Dear Dr. Gutierrez,

Thank you for submitting your manuscript to PLOS ONE. After careful consideration, we feel that it has merit but does not fully meet PLOS ONE’s publication criteria as it currently stands. Therefore, we invite you to submit a revised version of the manuscript that addresses the points raised during the review process. In particular, please address the suggestions by Reviewer 3. Please submit your revised manuscript by Jan 13 2024 11:59PM. If you will need more time than this to complete your revisions, please reply to this message or contact the journal office at plosone@plos.org. Please include the following items when submitting your revised manuscript:A rebuttal letter that responds to each point raised by the academic editor and reviewer(s). You should upload this letter as a separate file labeled 'Response to Reviewers'.A marked-up copy of your manuscript that highlights changes made to the original version. You should upload this as a separate file labeled 'Revised Manuscript with Track Changes'.An unmarked version of your revised paper without tracked changes. You should upload this as a separate file labeled 'Manuscript'.If applicable, we recommend that you deposit your laboratory protocols in protocols.io to enhance the reproducibility of your results. Protocols.io assigns your protocol its own identifier (DOI) so that it can be cited independently in the future. For instructions see: https://journals.plos.org/plosone/s/submission-guidelines#loc-laboratory-protocols. Additionally, PLOS ONE offers an option for publishing peer-reviewed Lab Protocol articles, which describe protocols hosted on protocols.io. Read more information on sharing protocols at https://plos.org/protocols?utm_medium=editorial-email&utm_source=authorletters&utm_campaign=protocols.

We look forward to receiving your revised manuscript.

Kind regards,

Siew Ann Cheong, Ph.D.

Academic Editor

PLOS ONE

Journal Requirements:

Reviewers' comments:

Reviewer's Responses to Questions

**Comments to the Author**

1. If the authors have adequately addressed your comments raised in a previous round of review and you feel that this manuscript is now acceptable for publication, you may indicate that here to bypass the “Comments to the Author” section, enter your conflict of interest statement in the “Confidential to Editor” section, and submit your "Accept" recommendation.

Reviewer #1: All comments have been addressed

Reviewer #3: (No Response)

2. Is the manuscript technically sound, and do the data support the conclusions?

Reviewer #1: Yes

Reviewer #3: Yes

3. Has the statistical analysis been performed appropriately and rigorously? 

Reviewer #1: Yes

Reviewer #3: Yes

4. Have the authors made all data underlying the findings in their manuscript fully available?

Reviewer #1: No

Reviewer #3: Yes

5. Is the manuscript presented in an intelligible fashion and written in standard English?

Reviewer #1: Yes

Reviewer #3: Yes

6. Review Comments to the Author

Reviewer #1: (No Response)

Reviewer #3: Authors sought to describe the sensitivity and specificity of symptoms and signs of COVID-19 for predicting the disease using a data set from their country. The is worth conducting and findings could improve on our knowledge in the subject matter.

Comments

Line 6: Authors should use numbers to denote authors names and not symbols

Line 59: Authors should amend COVID to COVID-19

Line 73: Authors should replace the statement "to be nadir in cases" with a more appropriate sentence

Line 84: Authors should indicate the software used for the analysis

Line 98: Authors should amend COVID-19,, to "COVID-19"

Line 102: Authors should correct the sentence "SARS-COV-2aespecially" and remove the bracket at the end of the sentence

Line 105: Authors should include the total numbers for each symptom and wave observed. This will help improve understanding of the data

Line 112: Authors should correct "high lev"

Line 136: Authors should remove "in" and correct the sentence.

Line 151: Authors should explain the full meaning of "main" under the table. The various symptom combinations should be indicated

Line 154: Authors should review literature on other studies which have describe the ROCs of these symptoms and other they compare to this current findings. The discussion needs more through review

General comments:

Authors should indicate if they received IRB approval to conduct this study.

7. PLOS authors have the option to publish the peer review history of their article (what does this mean?). If published, this will include your full peer review and any attached files.

Reviewer #1: No

Reviewer #3: No

---

## [Author Response · Author response to Decision Letter 2]

4 Dec 2023

Please see cover letter for the response

---

## [Editor Report · Decision Letter 3]

12 Dec 2023

How predictive of SARS-CoV-2 infection are clinical characteristics at presentation among individuals with COVID-like symptoms treated at the Mexican Institute of Social Security.

PONE-D-22-14771R3

Dear Dr. Gutierrez,

We’re pleased to inform you that your manuscript has been judged scientifically suitable for publication and will be formally accepted for publication once it meets all outstanding technical requirements.

Kind regards,

Siew Ann Cheong, Ph.D.

Academic Editor

PLOS ONE
---

## [Editor Report · Acceptance letter]

13 Dec 2023

PONE-D-22-14771R3 

PLOS ONE

Dear Dr. Gutierrez, 

I'm pleased to inform you that your manuscript has been deemed suitable for publication in PLOS ONE. Congratulations! Your manuscript is now being handed over to our production team.

Kind regards, 

on behalf of

Dr. Siew Ann Cheong 

Academic Editor

PLOS ONE